

# Storm-induced water dynamics and thermohaline structure at the tidewater Flade Isblink Glacier outlet to the Wandel Sea (NE Greenland)

Sergei Kirillov[1], Igor Dmitrenko[1], Søren Rysgaard[1,2], David Babb[1], Leif Toudal Pedersen[3], Jens Ehn[1],

Jørgen Bendtsen[2,4], David Barber[1]

[1] Centre for Earth Observation Science, University of Manitoba, Winnipeg, Canada
[2] Arctic Research Centre, Aarhus University, Aarhus, Denmark
[3] Danish Meteorological Institute, Copenhagen, Denmark
[4] ClimateLab, Copenhagen, Denmark

*Correspondence to:* Sergei Kirillov (sergei.kirillov@umanitoba.ca)

**Abstract.** In April 2015, an ice-tethered conductivity-temperature-depth (CTD) profiler and a down-looking Acoustic Doppler Current Profiler (ADCP) were deployed from the landfast ice near the tidewater glacier terminus of the Flade Isblink Glacier in the Wandel Sea, NE Greenland. The three week timeseries showed that water dynamics and the thermohaline structure were modified considerably during a storm event on 22-24 April when northerly winds exceeded 15

m/s. The storm initiated downwelling-like water dynamics characterized by on-shore water transport in the surface (0-40 m) layer and compensating off-shore flow at intermediate depths. After the storm, currents reversed in both layers, and the relaxation phase of downwelling lasted ~4 days. Although current velocities did not exceed 5 cm/s, the enhanced circulation during the storm caused cold turbid intrusions at 75-95 m depth that are likely attributed to sub-glacial water from the Flade Isblink Ice Cap. It was also found that the semidiurnal periodicities in the temperature and salinity time series were

associated with the lunar semidiurnal tidal flow. The vertical structure of tidal currents corresponded to the first baroclinic mode of the internal tide with a velocity minimum at ~40 m. The tidal ellipses rotate in opposite directions above and below this depth and cause a divergence of tidal flow which was observed to induce semidiurnal internal waves of about 3 m height at the front of the glacier terminus.

Our findings provide evidence that shelf-basin interaction and tidal forcing can potentialy modify coastal Wandel Sea waters

even though they are isolated from the atmosphere by landfast sea ice almost year round. The northerly storms over the continental slope cause an enhanced circulation facilitating a release of cold and turbid sub-glacial water to the shelf. The tidal flow may contribute to the removal of such water from the glacial terminus.

## 1. Introduction



The Arctic region has recently experienced a rapid rise of atmospheric temperature, a steep decrease of sea ice cover, and an
accelerated loss of glacial ice in Greenland. These changes are associated with the direct melting of snow, sea ice and
glaciers (i.e., Krabill et al., 2004; van den Broeke et al., 2009; Bamber et al., 2012; Shepherd et al. 2012) and calving of
tidewater glaciers (Reeh et al., 2001; Amundson et al, 2010; Rignot et al., 2010; Vieli and Nick, 2011). A description of the
coastal circulation is of great importance for better understanding these losses in the context of the oceanic heat transport
toward the glaciers, its impact on the rate of ice melting, and the melt water release to the open ocean. However, the water
dynamics in the coastal waters near the glaciers are not often well known. The ocean-glacier interaction is usually described
with a two-layer circulation scheme that consists of inflowing salty and warm ocean water at depth and outflow of a light
mixture of glacial meltwater and oceanic water at the surface (Straneo et al., 2011; Willson and Straneo, 2015). Such an
interaction is typical for the tidewater outlet glaciers located in relatively deep fjords surrounding the coast of Greenland
(Straneo et al., 2010), while the shelf-basin glacial meltwater exchange over the wide open shelves is determined by the local
coastal circulation which is often poorly known due to the lack of observational data in these remote regions.

The Wandel Sea (Figure 1a; also referred to as McKinley Sea) occupies the shelf area between cape Nordøstrundingen (81°
22' N, 11° 40' W) and Herluf Trolle Land (NE Greenland). It can be a significant source of freshwater that originates from
the summer snow/sea ice meltwater (Dmitrenko et al., 2017, submitted; Bendtsen et al., 2017) and glacial runoff (Willis et
al., 2015). The latter source might be of importance since numerous outlets of the Greenland Ice Sheet (in Independence and
Hagen fjords) and Flade Isblink Ice Cap (FIIC) drain meltwater into the Wandel Sea via three marine terminating glaciers
(Palmer et al., 2010). Despite the fact that the Wandel Sea is covered with landfast sea ice almost all year round, the fjords
and inner shelf have been mostly ice-free in August and early September during the last decade. These near-land open water
areas are separated from the deep ocean by a permanent multi-year landfast ice fringe (~3.5 m and thicker, Dmitrenko et al.,
2017 submitted) between FIIC to the northern tip of Prinsesse Margrethe Islands or even right up to the Peary Land coast.
This ice fringe remains in place throughout the year and probably traps and accumulates meltwater in the surface layer of the
inner part of the Wandel Sea. This effect seems to be rather similar to that reported by, e.g., Macdonald et al. (1995) and
Mueller et al. (2003), who suggested that the terrestrial waters at Mackenzie shelf and northern Ellesmere Island (Canadian
Arctic) were trapped behind landfast ice barriers that were thicker at the marine edge due to heavy ridging and *stamukhi*
formation. Results from the oceanographic survey carried out from the landfast ice in the Wandel Sea in April-May 2015
(Dmitrenko et al., 2017, submitted) support this hypothesis by revealing low salinities of 16-21 within 1.5-5 m in the surface
layer beneath the ice cover – much lower than one might expect for this region at the end of winter. For instance, the surface
salinity within the East Greenland Current off cape Nordøstrundingen remains above 31 year round (Rudels et al., 2005;
Jones et al., 2008; Kattner, 2009).

The draining of this freshwater reservoir to the open ocean is of importance when considering the potential perspective of a
seasonal disappearance of the landfast ice fringe under the climate warming scenario. However, even under present ice
covered conditions, coastal circulation can affect the shelf-to-basin freshwater flux through storm surges and associated
upwelling/downwelling, and vertical mixing. Tidal dynamics is another factor affecting the intensity of tidewater glacier



ablation (Mueller et al., 2012), the exchange of subglacial waters to the adjacent continental shelf (Makinson and Nicholls, 1999), and vertical mixing below the landfast ice cover (Dmitrenko et al., 2012). The role of tides is also evident in tidewater

glacier flexing and calving (e.g. Johnson et al., 2011; Rignot et al., 2010) that might further enhance the freshwater input to the coastal waters (Bartholomaus et al., 2013). Although Dmitrenko et al. (2017, submitted) demonstrated that frontal interleaving was occurring at the outer Wandel Sea shelf, none of the mentioned mechanisms was carefully considered due to the the scarcity of oceanographic observations in the Wandel Sea region.

In this study, we present a timeseries of CTD and velocity observations collected in spring (21 April to 11 May) 2015 from

the multiyear landfast ice in the front of the FIIC tidewater glacier outlet. We analyze the near-glacier current velocities and directions in relation to wind and tidal forcing, and relate the observed changes in vertical thermohaline structure to water dynamics.

The paper is organized as follows. The observational mooring data is described in section 2. In section 3, we provide an overview of the regional oceanographic setting and describe the thermohaline properties of water masses interacting with the

FIIC glacier outlet. Section 4.1 discusses the effect of a storm event on the temporal evolution of water dynamics and thermohaline structure near the glacier front. Then, section 4.2 focuses on the modification of the vertical thermohaline structure by tidal motions in the front of the glacier and provides evidence that the observed vertical water displacements at the glacial terminus resulted from a tidal flow divergence.

## 2. Data and methods

The oceanographic mooring was deployed from the landfast ice at the position 81.671°N 16.027°W, which is about 300 m off the terminus of the FIIC tidewater glacier (Figure 1). The water depth at the location was 110 m. The mooring setup consisted of (i) an ice tethered 300 kHz Workhorse Sentinel Acoustic Doppler current profiler (ADCP) by Teledyne RD Instruments placed at 2 m depth looking downward and (ii) an Ice Tethered Profiler (ITP) by McLane Research Laboratory equipped with a conductivity-temperature-depth (CTD) sensor 41CP by Sea-Bird Electronics and a Wetlabs ECO sensor for

measuring backscatter intensity (turbidity) at 700 nm, Chlorophyll (Chl) fluorescence, and colored dissolved organic matter (CDOM) fluorescence. The ITP was programed to cast every 2 hours in the depth range from 3 to 101 m with temperature and salinity recorded approximately every 0.3 m, while the backscatter intensity was recorded with a vertical resolution of ~1.5 m. The Chl and CDOM data obtained with ECO are not used in this paper. The inherent measurement errors associated with CTD sensor are small and do not exeed $2 \cdot 10^{-3}$ °C for temperature and $2 \cdot 10^{-3}$ for salinity according to the manufacturer.

The ADCP recorded water column current velocities averaged over 5 minutes with 4 m vertical intervals starting from 6 m depth (i.e., about 4 m below the ice cover). Due to low backscattering in the water column, the level of noise in the velocity data was relatively high. As a result the amount of reliable velocity data gradually decreased with depth with the attenuation of the signal strength. The automatic data processing routine provided by the manufacturer and based on signal correlation between the individual beams restrained the depth range with reliable data to the upper 66 m of the water column. The

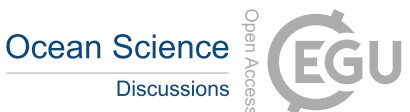

amount of data marked as "bad" in this range was less than 50%, and there were no "bad" records above 54 m. In the following, we therefore, only analyze current data obtained within the top 66 m. The accuracy of the ADCP current speed is 0.5% of the magnitude while the error in the compass direction is 2°. However, the small horizontal component of the Earth's magnetic field at the mooring location seemed to result in relatively large variations in ADCP heading. The heading direction varied randomly within a ~5° range, even though zero fluctuations are expected for the ice-tethered unit. Therefore, we conservatively estimate the total error of direction measurements as ±7° (i.e. 5°+2°). All currents were corrected for the local magnetic deviation (~18°W) taken from International Geomagnetic Reference Field model ([www.ngdc.noaa.gov/IAGA/vmod/home.html](www.ngdc.noaa.gov/IAGA/vmod/home.html)). Vertical velocity data from the ADCP was also used in the paper in order to demonstrate the amplitudes of baroclinic tidal waves at the front of glacier. The MATLAB Tidal Analysis Toolbox by R. Pawlowicz, R. Beardsley and S. Lentz was used to separate the tidal signal in temperature, salinity and current records from the non-tidal components (Pawlowicz et al., 2002). This routine is a translation of a FORTRAN package developed by Foreman (1977, 1978).

The mooring observations were complemented by manual CTD (SBE-19plus, Sea-Bird Electronics) observations obtained through auger holes drilled in the landfast ice in the period between 17 April and 15 May 2015 (Fig. 1b). The CTD accuracies in temperature and salinity are comparable to those for the ITP CTD sensor. Two oceanographic stations were carried out at the mooring position: #13 on 17 April (three days prior to the ITP deployment) and #84 on 5 May (following the recovery). In this research, we use the spatial CTD data in order to relate the T/S changes observed with the ITP to those measured in the vicinity of the FIIC glacier outlet. We also use several CTD profiles taken (i) in the southern part of the Wandel Sea over the continental slope between 204 m and 362 m depths during ARK-XXIII/2 expedition onboard *"R/V Polarstern"* in August 2008 (white circles in Fig. 1a), and (ii) in the vicinity of the FIIC terminus in August 2015 and April 2016 (pink circles in Fig. 1b). Note that CTD # 26, 27 and 28 were conducted from the sea ice within the debris of the fractured FIIC terminus during April 2016.

In situ obervations of wind speed and direction at 30 minute intervals were continuously recorded at Station Nord (Fig. 1a) at 9 m height above the ground, which corresponds to 46 m above the sea level. These recordings were accompanied by the 6-hour surface wind and SLP over NE Greenland obtained from the National Centers for Environmental Prediction (NCEP) reanalysis.

## 3. Regional oceanographic settings of the southern Wandel Sea shelf

The ocean circulation over the continental slope of the Wandel Sea is characterized by a general north-to-south directed water transport, in accordance with the general southward direction of the East Greenland Current (e.g. Aagaard and Coachman, 1968; Foldvik et al., 1988; Marnela et al., 2013). However, nothing is known about the circulation over the Wandel Sea shelf, which is covered with multiyear and first-year landfast sea ice almost year round. A general conception about the shelf circulation can be deduced from the analysis of the local thermohaline structure. Dmitrenko et al. (2017)



reported that the shelf waters of the Wandel Sea during winter are mainly comprised of Pacific water (PcW) in the depth range of 20-70 m and warmer Arctic-derived Atlantic water (ArAW) outflowing the Arctic Ocean via the western Fram Strait at greater depths. Within the cold halostad, represented by PcW, salinity increased from about 30 to 31.5 and

temperature decreased from -1.6 °C to -1.7 °C (Fig. 2a,b). The warm layer of relatively weakly stratified ArAW was observed at the Wandel Sea shelf between 100 and 185 m (the maximal depth sampled over the shelf – Fig. 1b). In this layer, the temperature and salinity increased from about -1°C to +0.3°C and from 34 to 35, respectively. The PcW and ArAW layers were separated by warm halocline water at 70-100 m, while the cold and extraordinarily low salinity waters in the uppermost 15-20 m layer were diluted by local freshwater sources (Dmitrenko et al., 2017). Bendtsen et al. (2017) reported

on a similar vertical thermohaline structure west of FIIC glacier observed during the short ice-free period in August 2016. The only changes occurred in the surface layer, where considerable freshening (S < 10) and warming (θ > +2°) were observed in relation to snow/ice melting and solar radiative heating.

The presence of PcW in the region seems to be limited predominantly to the shelf areas. In the deeper off-shelf areas, the vertical thermohaline structure is represented by cold Polar Water and ArAW. According to CTD profiles obtained over the

continental slope in 2008 (ARK-XXIII/2 expedition), the cold Polar Water with temperatures below -1.6 °C and salinities 32.0-34.0 is observed in the upper 110-120 m (Dmitrenko et al, 2017). Thus, underlying Arctic-derived Atlantic water is found considerably deeper over the continental slope than over the shelf area. Moreover, the lack of fresher PcW above ArAW results in the absence of the warm halocline layer over the continental slope (Dmitrenko et al., 2017).

The bottom topography of Wandel Sea adjacent to the northern FIIC glacier outlets is poorly known with only depth

measurements available from the sparse echo-sounding and CTD pressure records obtained in 2006 and 2015, respectively (Nørgaard-Pedersen et al., 2008; Dmitrenko et al., 2017). The key feature of bottom topography near the FIIC glacier is a submarine glacier valley with depths increasing from ~130 m (2 km east of mooring position) to more than 180 m measured about 18 km north of the glacier terminus (Fig.1b). One can surmise that this valley continues toward the continental slope thereby providing a gateway for the intermediate Atlantic Water inflow to the shallow areas, although other submarine

valleys could also exist in this area (Kattner, 2009).

## 4. Results and Discussion

### 4.1. The storm-induced intrusions of cold subglacial waters

The atmospheric forcing during the mooring deployment period was marked by a storm event associated with the development of a high atmospheric pressure anomaly over Greenland during 21-23 April (DOY 111-113, Fig. 3). The

NCEP-derived wind over the continental slope off the Wandel Sea shelf (82.5 °N, 10 °W) increased from 22 to 24 April, 2015 (DOY 112-114) and turned gradually from a westerly to a notherly direction (Fig. 4). The maximum wind speed of


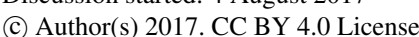


13.7 m/s was observed at 06:00, 24 April. At the regional scale, the wind speed over northeast Greenland exceeded 18 m/s on 24 April, although the extreme wind speeds might be underestimated by NCEP reanalysis (e.g. Jacobsen et al., 2012). In accordance with the NCEP reanalysis data the meteorological observations made at Station Nord also indicate strong

northerly winds during 22-24 April (Fig.4). The maximum wind speed of about 13.2 m/s (observed in 22 April, 20:30) is comparable to maximum NCEP-derived wind velocity over the continental slope, although it was observed ~1.4 days earlier and that might be attributed to the transition of the maximum windspeed core over NE Greenland (Fig.3).

The current velocities recorded in the front of the FIIC tidewater glacier terminus generally demonstrate very low dynamics. The average velocity over 6-66 m between 21 April and 11 May is only 0.3 cm/s (heading to T71°) with a standard deviation

of about 2.0 cm/s (Fig. 4c,d) that is mostly attributable to tidal dynamics. The storm passing over NE Greenland during 22-24 April resulted in current intensification in the upper 66 m and induced the development of a two-layer circulation cell that persisted from 23 to 30 April (DOY 113-120.5). This circulation clearly shows two different phases. The first one from 23-25 April (DOY 113.0-115.5, *event A*) was characterized by a west-southwest flow with a mean velocity of 0.7 cm/s within the 6-38 m surface layer, and a east-northeast flow from 46-66 m depth with a mean velocity of 0.9 cm/s (Fig. 4c,d). The

circulation cell reversed on 26 April such that east-northeastward flow in the upper layer (*event B*, DOY 116.5-120.5) became opposed by southwest flow below with respective velocities of 1.7 and 0.7 cm/s (Fig.4c,d).

Both the winds derived from NCEP and from the Station Nord weather station indicate downwelling-favorable wind conditions over the NE Greenland coast during 22-24 April 2015 (DOY 112-114). Although the consolidated pack ice over the continental slope is expected to strongly restrain the momentum flux, ice drift speed in the Fram Strait has been found to

be closely related to geostrophical wind speeds (Smedsrud et al., 2011). This implies that the stronger wind forcing will be transferred to the ocean surface to an extent proportional to the wind speed increase. Taking the observed wind pattern into account, one may suggest that the current pattern during *event A* was attributable to typical wind-driven downwelling dynamics associated with on-shore Ekman transport in the surface layer (6-38 m) and a compensating undercurrent at depth (46-66 m and, probably deeper). Following the storm and a reduction in wind speeds the current in both layers reversed for

the next 4 days (*event B*) during what we refer to as a relaxation period.

Progressive vectors were compiled for every depth of the ADCP records for better visualization of the pattern of residual (de-tided) currents during the storm event (*event A*) and relaxation period (*event B*) (Fig. 5). Despite relatively low residual current speeds, the apparent two-layer pattern is evident and the currents demonstrate predominately opposite directions relative to the 42 m level. A mean water transport on the order of 1 km per day was observed within the deep layer during

both events, whereas the mean speed in the surface layer increased from ~1 km to ~2 km per day in the period between *events A* and *B*. The trajectories of parcels also consistently indicate water transport heading E-NE and W-SW, which reasonably corresponds to the orientation of the glacier terminus (Fig. 1b).

Although the observed horizontal displacements during the storm event and the following relaxation period were relatively small and comprised of 2-3 km in the deep layer, some remarkable changes of the vertical thermohaline structure in the ITP

profiles are evident below 70 m in 23-30 April (DOY 113-120.5). These changes appear in the form of thermohaline




intrusions that are identified as local extrema ("spikes") in the TS space within the salinity range of 32.0-33.8 (Fig. 6). These intrusions occupied the ~75-95 m layer that corresponded to warm halocline waters over the shelf region. The presence of intrusions usually characterizes the interleaving of one water mass into another along isopycnal surfaces in frontal zones (e.g., Walsh and Carmack, 2003; Rudels et al., 2005; Woodgate et al., 2007). In the Arctic, the interleaving typically occurs

along the quasi-isohaline surfaces due to the small thermal expansion coefficient of sea water at low temperatures. The intrusions observed at the front of FIIC glacier were 0.2-0.4°C colder than the ambient waters (Fig. 6c), while the mean water temperatures in the range of 80-101 m near the glacier terminus were on average 0.07-0.08°C colder than ambient waters at the same depths (see yellow line and white circles in Fig. 6c).

In contrast to the well-pronounced intrusions in front of the glacier, the CTD profiles from ambient stations show much

smaller intrusive "spikes" over the Wandel Sea shelf (Fig. 6c). Dmitrenko et al. (2017) attributed the thermohaline intrusions over the *outer* Wandel Sea shelf to the lateral interaction between the on-shelf warm halocline waters and the off-shelf cold Polar Water. The reference CTD profiles (ARK 267, 268, and 270, Fig. 1a) obtained over the continental slope in August 2008 generally support the idea that the thermohaline intrusions in front of the glacier (in the ITP profiles) also can be caused by such interaction. For instance, the temperatures at 33.2 isohaline (that roughly approximates the center of largest

intrusions in the TS space) at the reference CTD stations match those in the largest intrusions observed at the ITP site (Fig. 6c). However, the relatively small horizontal displacements of water masses during both episodes would imply the presence of a very narrow (2-3 km wide) frontal zone dividing the warm halocline and off-shelf cold Polar Water in close vicinity to the mooring position. Also, the temperature and salinity profiles at station #74 (located ~3 km further towards the continental slope from the mooring position) show no presence of cold Polar water at the depth of the intrusive layers. The temperature

corresponding to the 33.2 isohaline at station #74 was above -1.32°C (not shown), whereas the observed thermohaline intrusions from ITP observations bear much colder waters with temperatures as low as -1.7°C (Fig. 6c).

Considering the temperature and salinity profiles obtained at stations #13, 72, 73, and 75, Dmitrenko et al. (2017) suggested that relatively small cold intrusions of turbid water in the vicinity of the FIIC glacier observed at ~87 m could be attributed to subglacial waters enriched with suspended matter. Based on the Archimedes principle and an approximate terminus height

of 10 m above sea level, we estimate that these intrusions roughly correspond to the theoretical depth of the tidewater terminus of the FIIC glacier.

In August 2015, the detailed oceanographic survey accomplished over the ice-free area west of the FIIC glacier tongue showed the absence of warm halocline water in the vicinity of the glacier. Based on CTD data from August 2015, Bendtsen et al. (2017) argued that bottom water from the shelf can intrude below the tidewater glacier and cool to the freezing

temperature through heat loss to the glacial ice. This cooling is not accompanied by basal melting and, hence, the salinities of underlying water below the glacier are not changed. Temperature and salinity near the bottom of their deepest stations near the glacier terminus were -1.73°C and 33.02, respectively (st . 166, 70 m, Fig. 1b), which are reasonably close to the characteristics of the observed intrusions from the ITP (Fig. 7). Moreover, several CTD stations carried out within the debris of the glacier terminus in April 2016 (stations #26-28, Fig. 1b) show water temperatures between -1.67 and -1.74°C at the



33.2 isohaline. The TS profiles from these reference stations are shown in Fig.7 along with scatterplots of all ITP data obtained during *event A* and *B*.

Although the water dynamics at the depths of intrusions is unresolved with ADCP data, we speculate that along-glacier water transport at 75-95 m is similar to that found in the lower layer of the circulation cell (i.e. at 46-66 m). This implies that the intrusive layering is accompanied by off-shelf currents during the storm event and on-shelf water transport during the

period of relaxation. The structure of intrusions was different in *events A* and *B*. During *event B*, relatively colder and more turbid (with higher optical backscatter intensities) intrusions were evident compared to *event A* (Fig. 7). Moreover, the interleaving observed during the relaxation period (*event B*) was, in fact, presented by the single intrusion of cold water ($\theta$ = -1.7°C) gradually deepening over time (Fig. 8). The fractured character of the calving terminus of the FIIC, a lack of depth measurements along the terminus, and the generally unknown spatial pattern of glacier draft do not allow us to draw a

conclusion of why the opposite currents caused the observed differences in the sub-glacial water intrusions. However, we suggest that it is the along-terminus flow impetus that created the frontal instability resulting in release of cold sub-glacial waters and their interleaving into the ambient water masses. The proximity of observed temperatures and salinities within intrusions to the thermohaline properties of cold sub-glacial waters at intermediate depths convincingly supports this suggestion. Moreover, there is no evidence of distinct interleaving intrusions beyond the period associated with downwelling

winds and relaxation flow (Fig. 7a,b, grey dots).

**4.2. The internal $M_2$ tide at the front of glacier**

In section 4.1, we mentioned that tidal flow contributes to the flow variability at the ITP site. The most energetic tidal currents are associated with the lunar semidiurnal constituent ($M_2$), whereas the current velocities associated with the next two principal harmonic ($K_1$ and $O_1$) are about 2-3 times smaller (not shown). The $M_2$ fluctuations were most pronounced in

the surface (1.9 cm/s) and deep (>60 m, 1 cm/s) layers at the front of glacier (Fig. 9a). The tidal ellipses were strongly elongated and mainly directed in the north-east to east directions (Fig. 9a). At mid-depths (30-42 m), the tidal velocities were relatively low with the minor axis changing from positive to negative values at 42 m which indicates a transition from counterclockwise (CCW) tidal current rotation in the surface layer to clockwise (CW) rotation at depth.

The vertical profiles of temperature and salinity tidal ($M_2$) amplitudes were, in general, similar and showed the relatively

large amplitudes within (i) the upper 10 m (amplitudes are up to 0.02°C and 0.34 for temperature and salinity, respectively) and (ii) at 70-90 m (0.04°C and 0.13, Fig. 9b). Within the mid depths, however, the $M_2$ amplitudes were small and did not exceed 0.01°C and 0.05, although both amplitudes were significant at 95% confidence level (Fig. 9b). The observed tidal T/S oscillations cannot be a result of lateral motions in the frontal area, since it would suggest the existence of a temperature and salinity front that was not confirmed by observations. For instance, the horizontal gradients of temperature and salinity

estimated from their tidal amplitudes and tidal currents within the 62-70 m layer were of 0.28° and 2.67 per kilometer, respectively. These estimates are about two orders of magnitude larger than the inherent horizontal T/S gradients within the





ambient warm halocline waters (not shown). The same differences can be reported for the surface 6-26 m layer, where horizontal T/S gradients of 0.30° and 4.40 per kilometer, estimated from the tidal ellipses' characteristics are considerably higher than the full range of spatial variability of temperature and salinity obtained from the entire CTD survey.

These discrepancies imply that the observed tidal fluctuations in T/S records presumably originated from a vertical displacement of isotherms and isohalines associated with the internal tide near the glacial terminus. Taking into account the vertical gradients of temperature and salinity evaluated from their mean profiles (Fig. 6a,b) one could expect that these displacements would cause out of phase T/S oscillations in the upper 60 m and the coherent T/S oscillations below. This is in fairly close agreement with observations (Fig. 9c). Using the mean vertical T/S gradients, we calculated the vertical

displacements that would result in the observed tidal oscillations in temperature and salinity at different depths. It was found that both temperature and salinity gave comparable displacements showing the largest amplitude (3 m) within 40-60 m layer and a gradual decrease toward the surface and bottom (Fig. 9d). The shape of displacement profiles corresponds to the first baroclinic mode of internal tide, which would also suggest maximum amplitude of vertical speeds and displacements at intermediate depths (e.g., Gill, 1982). Although the vertical components of current velocities were very noisy and the only

significant (at 95% level) $M_2$ amplitudes were obtained within 42-58 m, the displacement derived from the vertical speed (of ~4 m) is reasonably close to the 3-meter displacement found from T/S data (Fig. 9d).

The observed internal tide near the glacier terminus is likely explained by the divergence of the contrary rotating lunar semidiurnal currents above and below 42 m (CCW and CW, respectively, Fig. 10a). To prove this hypothesis, we estimated the amplitude of vertical displacement in the mooring position by considering the continuity of mass transport toward the

glacier in the upper layer ($D_1$, Fig. 10b) and off the terminus near the bottom ($D_2$) during one half of the tidal cycle (6.21 hours) – Fig. 10c. The assumption that the tidal ellipse below 70 m has the same amplitude and inclination as in the layer above was made to avoid neglecting the water mass transport within the bottom layer that was unresolved with velocity measurements. In order to determine the projection at which the absolute values of $D_1$ and $D_2$ are equal we tested different orientations of glacier terminus and found that projection on T343° gave the best correspondence between the on- and off-

glacier water transport (Fig. 10b). This projection corresponds reasonably well to the actual E-NE orientation of the glacier terminus and the main direction of the residual currents (Fig. 4). The typical scale of the vertical displacements at intermediate depths associated with the tidal divergence can be estimated from the following simple formulae:

$$O(h) = D_1/L$$

where $D_1 = \int_{42m}^{3m} \int_0^{6.21h} U \, dt \, dz = 1350 \, m^2$ is the intergrated water transport in the surface 42 m layer during one half of the tidal cycle, $U$ is the velocity projection on T343°, $L$ is the distance from the mooring position to the glacier (Fig. 10c).

The latter is assumed to be about 300 meters as determined from satellite image. However, the order of vertical displacements of 4.5 m obtained through this primitive approach agrees reasonably well with the results shown in Fig. 8. The discrepancy could be attributed to many factors that were not considered in our simple estimates. These might include the



unresolved bottom layer in the ADCP records, the horizontal inhomogeneity of vertical displacements (which are probably higher near the terminus), and the dissected and fractured nature of the glacier terminus.

**5. Conclusions**

The multiyear and firstyear landfast sea ice present in Wandel Sea insulates the Wandel Sea shelf from the wind forcing almost all year round. As a result, the momentum flux from the atmosphere to ocean is suppressed, and the local water dynamics is extremely weak during the ice-covered period. Although the average current speed in the front of FIIC tidewater glacier is less than 1 cm/s throughout the upper 70 m of the water column (with no reliable records at deeper levels), the

downwelling-favourable winds during a storm in April 2015 considerably modified the vertical structure of currents and amplified the current velocities. Northerly winds exceeding 18 m/s over the NE Greenland continental slope generated a two-layer circulation cell during 22-24 April, 2015 similar to what characterizes a typical downwelling situation. The on-shore water transport along the glacier terminus in the surface layer (0–42 m depth) was accompanied with a compensating off-shore flow below. During the storm event, the current velocities increased up to 4 cm/s in both layers and the turbid cold

water intrusions characterized by the negative temperature anomalies of up to 0.4°C appeared at 75-95 m depth. Similar intrusions were formed during the relaxation phase of the downwelling event between 26 and 30 April and after the wind abated. This relaxation phase was characterized by a reversal in current directions, with off-shore flow in the surface layer and on-shore flow at intermediate depths.

We suggest that the cold intrusions observed during 22-30 April 2015 off the FIIC glacier terminus have sub-glacial origin.

This suggestion is supported by data on vertical thermohaline structure obtained in the close vicinity of the glacier terminus and within the terminus debris in August 2016 and April 2016. Both surveys demonstrated colder temperatures within 75-95 m layer that were in contrast to the warmer Atlantic-modified halocline waters observed at depths off the terminus. Moreover, the depths of the intensive interleaving presumably corresponded to the draft of the glacier outlet that was roughly estimated from the height of the terminus (~8-10 m). Although the residual currents indicate predominantly along-terminus

water transport, at least above 70 m, we suggest that the observed intrusions occurred due to the impetus received from the enhanced circulation at the glacier terminus during the storm event.

The release of the sub-glacial cold water is not the only remarkable consequence of the storm-driven circulation over the Wandel Sea shelf. It was demonstrated that the on-shore water transport with residual currents in the surface layer during the storm event (~2 km) was about 3 times less than the off-shore water transport during the period of relaxation. Such

asymmetry of the net water transport might be considered as a potential mechanism that facilitates the release of freshwater accumulated under the landfast sea ice in the Wandel Sea shelf, to the open ocean, although a detailed study of this mechanism is beyond the scope of this paper.

Our results demonstrate that the tidal currents below the landfast ice at the FIIC terminus are mainly represented by lunar semidiurnal motions with the current amplitudes of up to 2 cm/s near the surface. The subsurface minimum of the $M_2$ current

amplitudes and the transition of $M_2$ tidal ellipse rotation from CCW in the surface layer to CW at depth correspond to the first baroclinic mode of the internal tide. It was revealed that the divergence of tidal flow at the glacier terminus leads to vertical displacements that also were evident in tidal oscillations in temperature and salinity. Although the amplitudes of vertical displacement were relatively small (up to 3 m at middle depths), the occurrence of internal tidal waves confirms the baroclinic origin of the tidal flow near the glacier terminus. This tidal-induced flow might maintain a slow but persistent

renewal of the sub-glacial waters taking into account the generally weak local water dynamics.

The basic importance of the current study is the fact that the wind-driven circulation and the tidal flow can considerably affect the Wandel Sea shelf waters during winter, regardless of the landfast ice cover, and impact the processes of shelf-glacier interaction and cold subglacial water release. This leads us to question the residence time of subglacial waters under the FIIC tongue, however, addressing this question will require observations that are geographically more focused on the

sub-glacial water properties and dynamics.

**Acknowledgments**

We gratefully acknowledge the support by the Canada Excellence Research Chair (CERC) and the Canada Research Chairs (CRC) programs, the Canada Foundation of Innovation (CFI), the Manitoba Research and Innovation Fund, the University of Manitoba, Aarhus University, the Greenland Institute of Natural Resources, and the EU project NACLIM,

grant agreement 308299. This work is a contribution to the joint Canadian-Danish-Greenland ASP cooperation. The research was also partly supported by National Sciences and Engineering Research Council of Canada (NSERC, grant RGPIN-2014-03606).

Authors strongly appreciate the efforts of Kunuk Lennert, Ivali Lennert and Egon Randa Frandsen for logistic support and their assistance in deployment and recovery of oceanographic equipment. We would also thank the Villum Research Station

for an excellent stay and the St Nord 'Fupper' for their hospitality.

The CTD and velocity data are archived in the Centre for Earth Observation Science (University of Manitoba) and are restricted for open access during two years after the observations completed. Afterward they will be posted on the ASP webpage at http://www.asp-net.org.

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

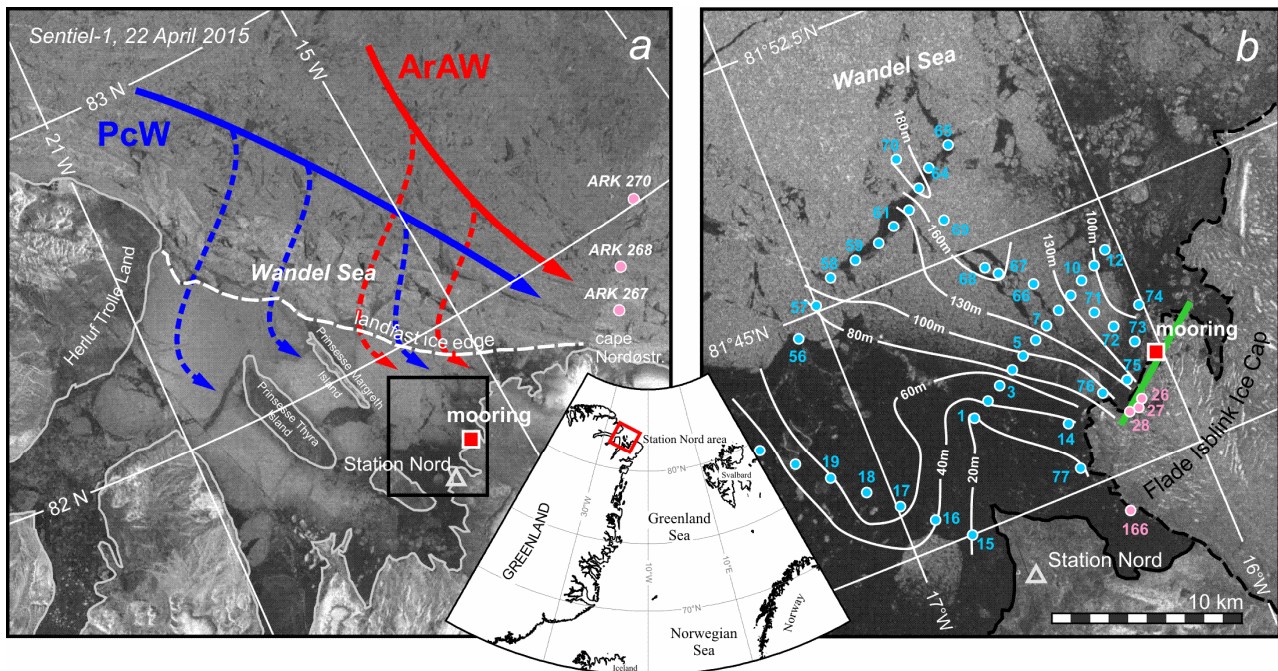

**Figure 1. (a) The Wandel Sea region and (b) the bathymetry map near the Flade Isblink Ice Cap outlet shown for the area demarcated by the black rectangle in (a). The red and blue arrows in (a) schematically indicate two major regional flows: the Arctic-derived Atlantic water (ArAW) and the Pacific water (PcW). The white contour lines show the bottom depth whereas the**

**black solid and dashed lines correspond to the coastline and glacier edge, respectively. Blue circles correspond to position of CTD**



stations in April-May, 2015, while pink circles indicate the positions of supplemental CTD profiles conducted in August 2008 (ARK 267, 268, and 270), in August 2015 (st. 166), and in April 2016 (st. 26, 27, and 28). The red squares show the location of the ice-tethered mooring (21 April – 10 May, 2015) equipped with the ITP and ADCP. The green line indicates the approximate orientation of the glacier terminus in vicinity of the mooring position.


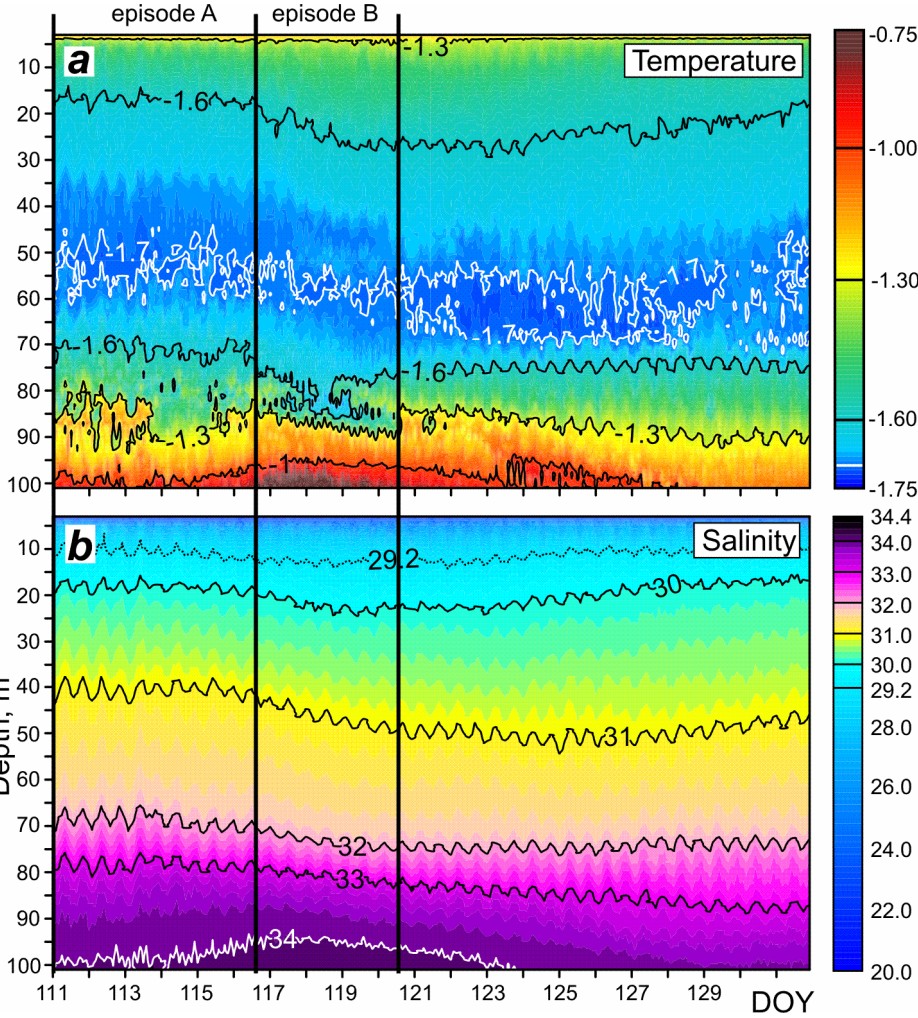

**Figure 2.** The temporal evolution of (a) temperature and (b) salinity measured by the Ice-Tethered Profiler (ITP) between 21 April, 00:00UTC and 11 May, 21:00UTC at the FIIC glacier terminus. The horizontal axis shows day of year (DOY).



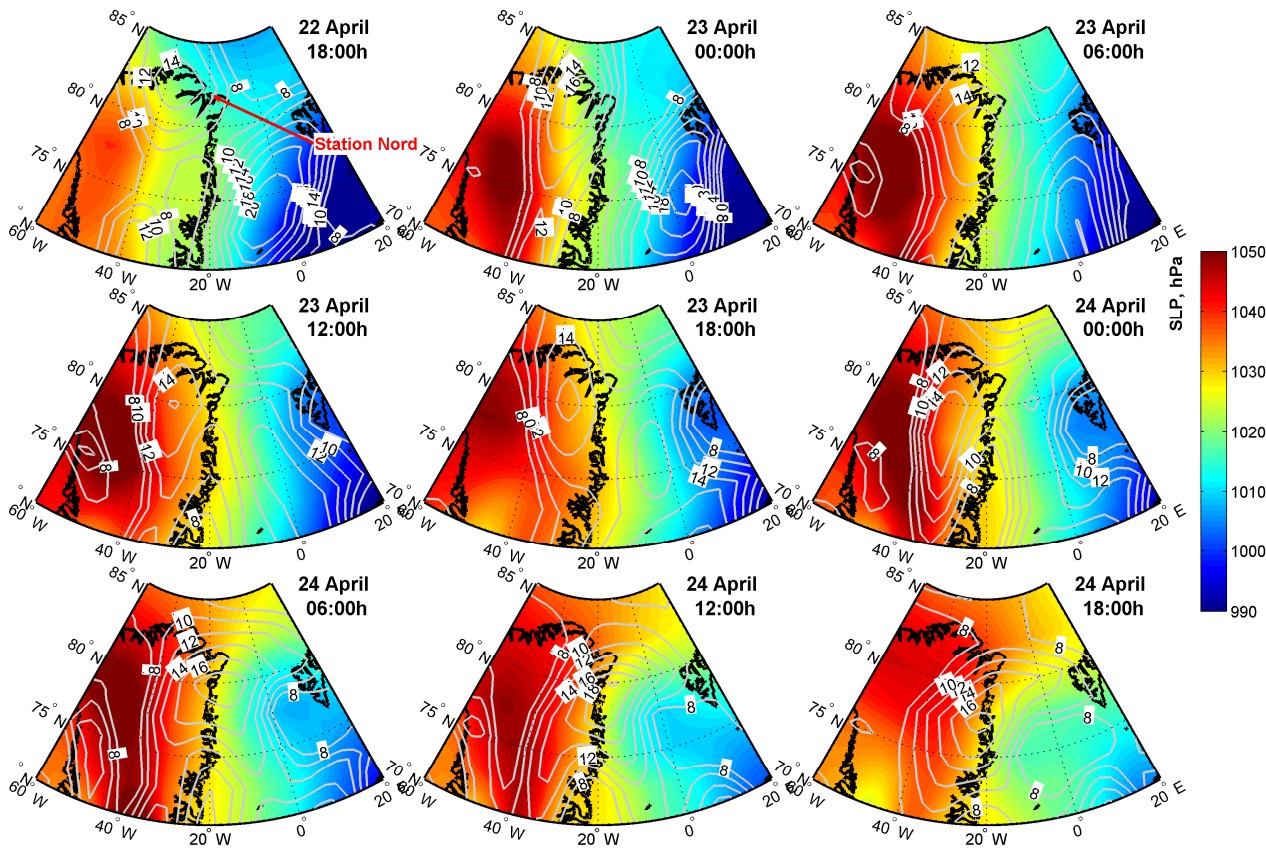


**Figure 3. The evolution of the NCEP-derived sea-level pressure (in colour, hPa) over Northeast Greenland during 22-24 April 2015. The grey numbered contours indicate the surface wind speed equally spaced between 8 and 18 m/s with 2 m/s interval.**





Figure 4. The temporal evolution of 2-h mean (a) zonal and (b) meridional velocities (cm/s) recorded between 21 April, 00:00h and 11 May, 21:00h with (c,d) their mean (solid black lines) and standard deviations (dashed black lines). Red and blue shading corresponds to positive and negative velocities, respectively. The red and blue lines correspond to mean velocities measured during episode A and B, respectively. The grey shading in (c) shows the percentage of "good" current records. Blank areas in (a) and (b) mask the records with less than 50% of good data in daily records. The lower panel demonstrates (top) the 6-h NCEP wind over continental slope at 82.5°N, 10.0°W and (bottom) 30-min wind measured at Station Nord.

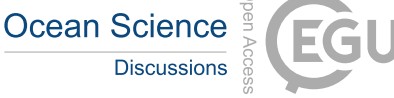

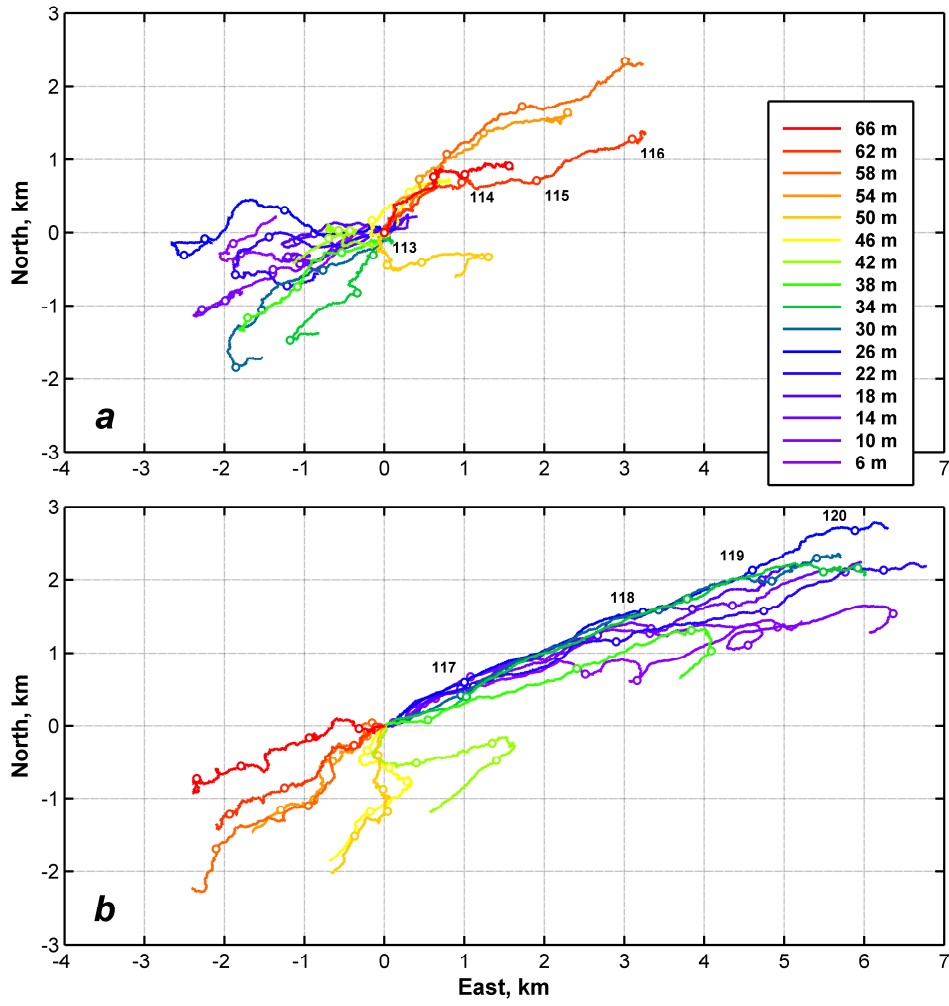

**Figure 5. The progressive vector diagram showing the residual (de-tided) currents during events (a) *A* (DOY 113.0-116.5) and (b) *B* (DOY 116.5-120.5). Circles indicate every DOY day at 00:00h.**




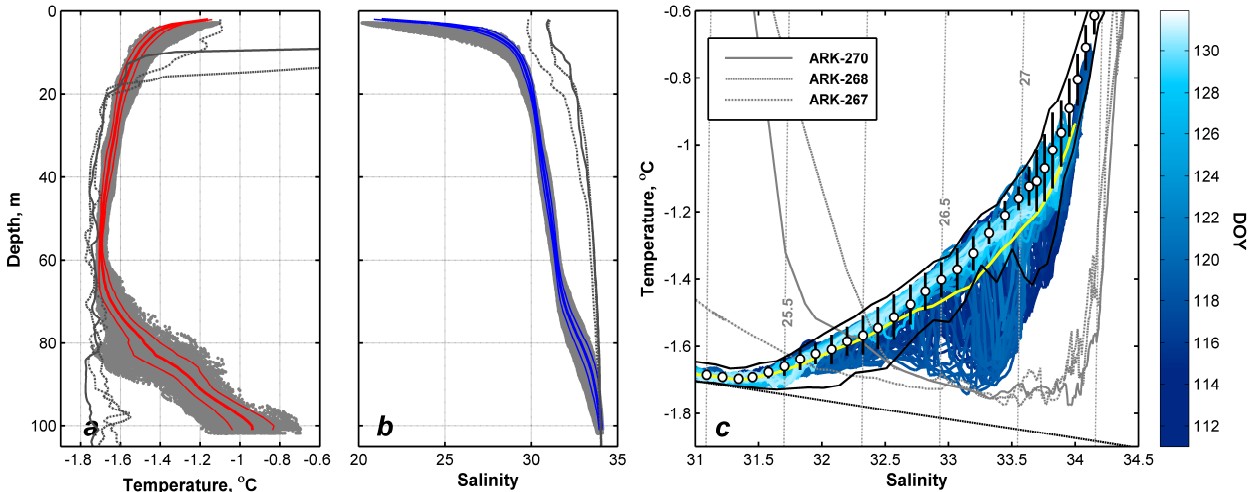

**Figure 6.** Scatterplot of all (a) temperatures and (b) salinities measured with the ITP. Red and blue lines indicate the mean and ± one standard deviation depicted with bold and thin lines, respectively. The grey curves show profiles measured on board R/V *Polarstern* (ARK 267, ARK 268 and ARK 270, see Fig. 1a) in August 2008. (c) TS diagram of all ITP profiles (blue curves) and their mean TS approximation (yellow line). White circles with vertical bars show the mean TS curve and standard deviation of temperatures for the ambient CTD profiles and the black lines encircle all TS data from ambient waters, both adopted from *Dmitrenko et al.* (2017). The dates of each TS-profile (in yeardays) are indicated by the color scale. Dashed gray lines are σ-zero isopycnals in kg m$^{-3}$, while the dashed black line is the freezing line.

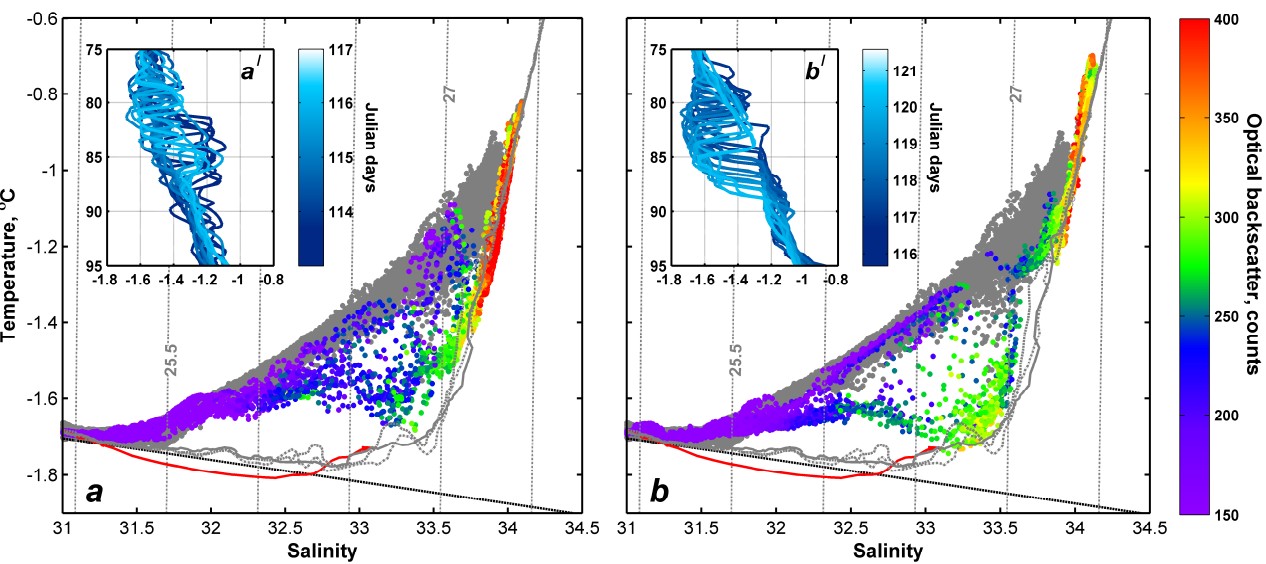





**Figure 7. The TS scatter-plot for events (a) *A* and (b) *B* when the active interleaving was observed (see Fig. 2) with optical backscatter intensity (color bar). The grey circles correspond to the TS measured after 30 April, 12:00UTC (DOY 120.5). The red and grey curve(s) show the TS profile(s) measured near the glacier in August 2015 (#166) and within the debris of glacier terminus**

**in April 2016 (#26, 27, and 28), respectfully. The inserts a' and b' show the subset of temperature profiles during each episode. Dashed gray lines are σ-zero isopycnals in kg m⁻³.**

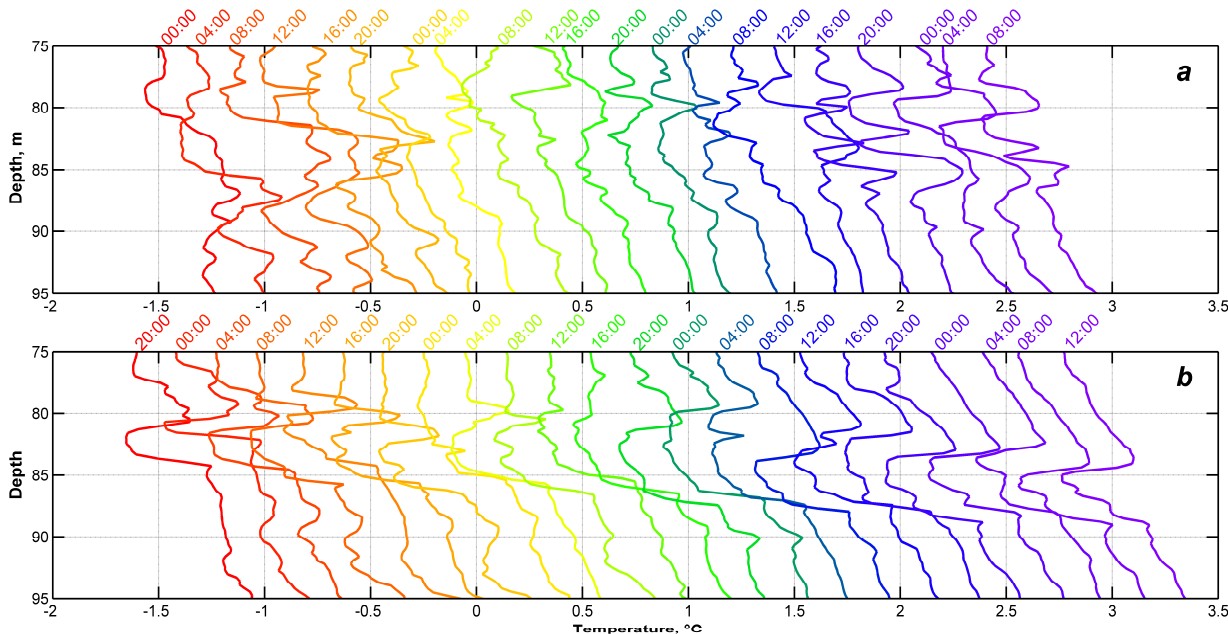

**Figure 8. The evolution of intrusive layers within 75-95 m depth during *events* (a) *A* (DOY 113-116.5) and (b) *B* (DOY 116.5-120).**

**Each profile is offset 0.2°C from its neighbor starting from first profile.**




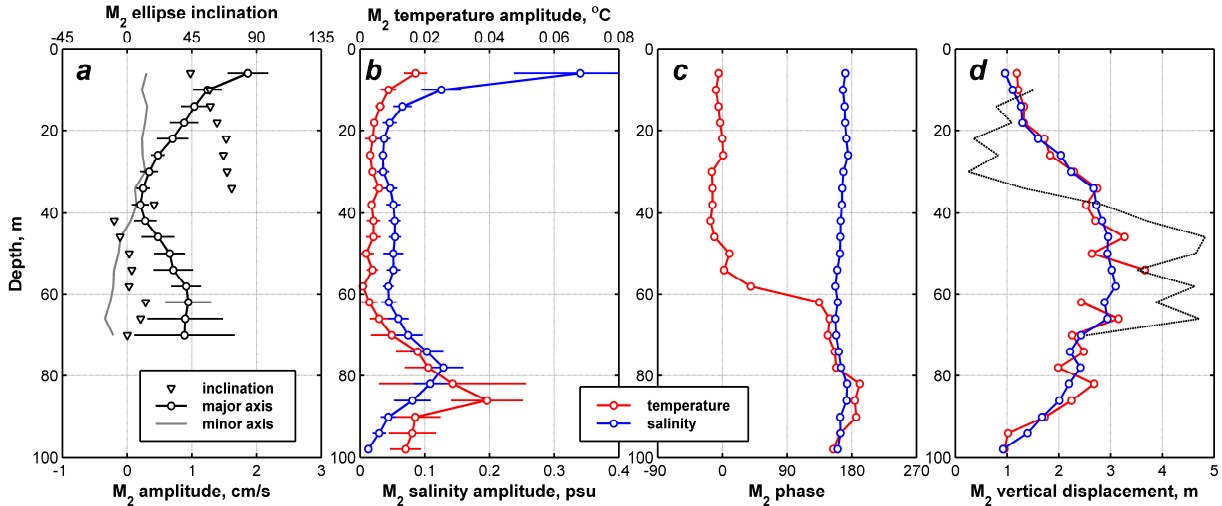

Figure 9. (a) Vertical distribution of the lunar semidiurnal ($M_2$) tidal ellipse parameters at the mooring position. Horizontal bars indicate the 95% confidence interval for major axis amplitude. (b) The tidal ($M_2$) amplitude of temperature (red) and salinity (blue) and the corresponding 95% confidence intervals. (c) The $M_2$ phases of temperature and salinity. (d) The vertical displacement (m) of water parcel estimated as a ratio of $M_2$ tidal amplitude of temperature/salinity (red/blue) to the vertical gradients of temperature/salinity estimated from the mean vertical profiles in Figure 5a and 5b. Dotted black line shows the vertical displacements estimated from lunar semidiurnal component of vertical velocity.

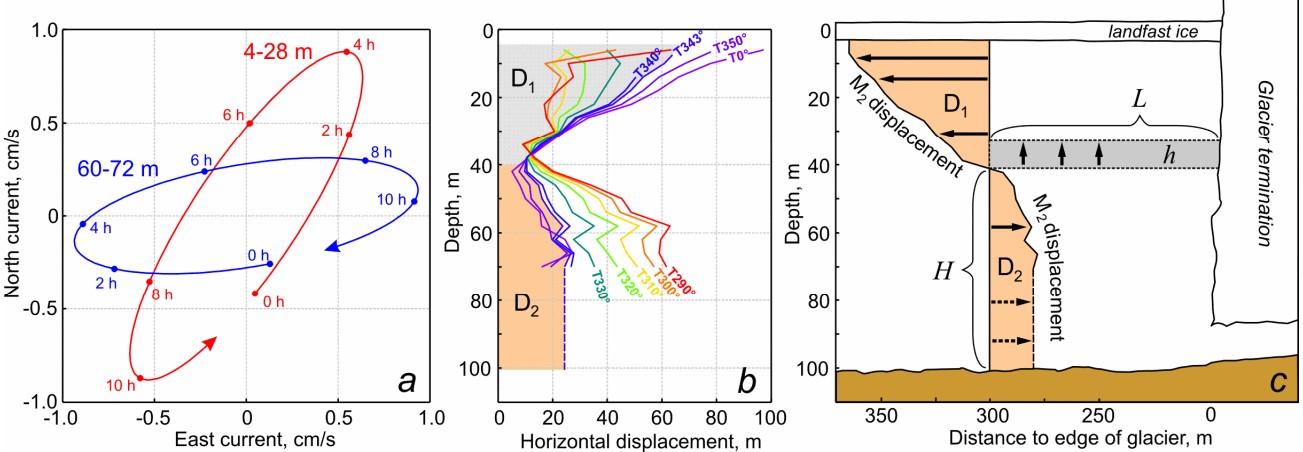



**Figure 10. (a) The mean lunar semidiurnal (M$_2$) tidal ellipses in the surface (6-26 m, red) and intermediate (62-70 m, blue) layers. Arrows show the rotation direction and dots indicate every 2-hour interval. (b) The absolute horizontal displacements of water parcels through the water column within a half M$_2$ tidal cycle (~6.21 hours) for different projections are shown with color lines (the numbers correspond to the projection angle from the North). The coloured numbers indicate the projection angles. The**
**shaded areas (D$_1$ and D$_2$, m$^2$; where D$_1$ = D$_2$) correspond to the total tidal transport in the surface (4-40 m) and bottom (40-101 m) layers projected at T343°. (c) The mechanism of the tidal internal wave generation at the glacier terminus associated with the divergence of the baroclinic M$_2$ tide.**