# Peer review of "Storm-induced water dynamics and thermohaline structure at the tidewater Flade Isblink Glacier outlet to the Wandel Sea (NE Greenland)"

_Ocean Science, 2017_

## Referee Comment (RC1) · Anonymous Referee #1 · 6 Sep 2017

I'll keep it short: this paper is pretty good, and the results are exciting. The manuscript shows that the water masses interacting with a tidewater glacier can be affected by storm events despite fast ice cover. It highlights that tides and storms can induce circulation in the vicinity of supposedly sheltered glaciers, which has dramatic implications for the future of these floating glaciers.

The science is rigorous, and since it is based on in-situ measurements I cannot require more data. There is however one thing that is missing: the explanations as to why you do each calculations that are demonstrated. Try and justify what you are doing, and link each paragraph and section a bit more together. As it is now, the paper is quite

dry and sometimes hard to follow. Help the reader understand where you are leading them.

Also, five minor comments:

- line 129 (p5): halostad? This is the first time I see this term.

- line 220 (p7): why is there no basal melting? The intrusions sound less problematic then.

- Figure 4, lower panels: have vertical lines over it to indicate the days, or at least the red and blue boxes of episodes A and B.

- Figure 6: in the caption, the distinction between what is measured from PS and from ITPs is unclear - everything looks like grey curves to me. Choose different symbols or a different wording.

- Figure 7: ylabel of the insets is missing

I consider that all these suggestions are easy to address and hence recommend that this article is accepted once my comments have been addressed.

---

## Referee Comment (RC2) · Anonymous Referee #2 · 28 Sep 2017

I find the manuscript mostly clear and with interesting results. I have only one request; the color coding of DOY (Day Of Year) in Figure 6c is not visualizing the spikes in temperature very well. It would be nice if the authord could change that, perhaps emphasize the A and B storm events. This goes for the inserts in Figure 7 as well.

---

## Author Comment (AC2) · 6 Oct 2017

We highly appreciate the reviewer's comments and have tried to do the best to satisfy his/her request about the color scale in Fig. 6c and Fig. 7. However, the initially chosen pattern seems to represent one of the best visualization of data we could find. Unfortunately, using more saturated patterns (i.e. having wider color spectrum) sometimes results in very odd visual effects. The example of such color scale can be seen in figure 1 attached here.

The reason why it is difficult to find the optimal color scale is a large amount of vertical profiles presented in Fig. 6 and Fig. 7. It is nearly impossible to find the color scale

that would be able to emphasize the different periods of the storm event. In fact, these figures aim to demonstrate the whole scope of TS or vertical profiles rather than the individual intrusions. The latter are shown separately in Fig. 8 where individual spikes in temperature are traced for different periods of storm.

Although we suggest that the initial color scale is more or less suitable, we have tried to find an alternative variant that would demonstrate the individual spikes more clearly. The figures with a new color scale are attached here as figure 2 and figure 3.
* * *
[Figure]

Depth, m
Temperature, °C
Salinity
a
b
c
ARK-270
ARK-268
ARK-267
25.5
27
DOY

**Fig. 1.**

[Figure]

a
b
c
Depth, m
Temperature, °C
Salinity
DOY
ARK-270
ARK-268
ARK-267
25.5
27
0
20
40
60
80
100
-1.8
-1.6
-1.4
-1.2
-1
-0.8
-0.6
20
25
30
35
31
31.5
32
32.5
33
33.5
34
34.5
112
114
116
118
120
122
124
126
128
130

**Fig. 2.**

[Figure]

Temperature, °C
Salinity
Depth, m
Temperature, °C
DOY
Optical backscatter, counts
a
b
a'
b'
25.5
27

**Fig. 3.**

---

## Author Response (AR2)

**Topic Editor Decision: Publish subject to technical corrections**

Line 119. "Centre's" or "Centre" (should be singular I think).

"Centre" was chosen

Line 229. Maybe "supports" rather than "sustains"?

We agree. The line has been changed.

Line 252. "harmonics"

Thank you. Changed accordingly.

Line 279. "metre" (preferable spelling) or maybe just "m".

Done.

Lines 307, 308. Better ". . and turbid . . by negative . ."

Done.

Line 315. "at depths". Do you mean "at similar depths" or "at greater depths"?

Thank you for this question. "At similar depth" is better here.

Line 327. Better ". . with current . ."

Done.

**Interactive comment by anonymous referee #1**

"I'll keep it short: this paper is pretty good, and the results are exciting. The manuscript shows that the water masses interacting with a tidewater glacier can be affected by storm events despite fast ice cover. It highlights that tides and storms can induce circulation in the vicinity of supposedly sheltered glaciers, which has dramatic implications for the future of these floating glaciers.

The science is rigorous, and since it is based on in-situ measurements I cannot require more data. There is however one thing that is missing: the explanations as to why youdo each calculations that are demonstrated. Try and justify what you are doing, and link each paragraph and section a bit more together. As it is now, the paper is quite dry and sometimes hard to follow. Help the reader understand where you are leading them."

**Response to interactive comment of anonymous referee #1**

We appreciate the evaluation of our paper by the first referee. In order to highlight our individual findings and link them together we modified the original text according to the reviewer's recomendations. Specifically, the reviewer asked us to explain why we do each calculation and justify what we are doing. Although the reviewer has not specified which paragraphs (or lines) are hard to follow, we found several places in *Results and Discussion* where some improvements would be needed:

Line 164-168: We changed the beginning of paragraph:

"The storm passing over NE Greenland during 22-24 April considerably modified the generally low water dynamics in the front of the FIIC glacier terminus. The mean current speed over 6-66 m depth during the entire measuring period was only 0.3 cm/s (heading to T71°) with a standard deviation of about 2.0 cm/s (Fig. 4c,d) which is mostly attributable to tidal dynamics. The storm resulted in current intensification and induced the development of a two-layer circulation cell that persisted from 23 to 30 April (DOY 113-120.5)."

Line 187-188: We changed the sentence "*Progressive vectors were compiled for every depth of the ADCP records for better visualization of the pattern of residual (de-tided) currents during the storm event (event A) and relaxation period (event B) (Fig. 5).*" with a new one:

"To demonstrate the horizontal extent of water transport during the storm (event A) and relaxation period (event B) we compiled the Lagrangian paths of residual (de-tided) currents for every depth of the ADCP records (Fig. 5)."

Line 207-208: A new sentence was added:

"The recorded intrusive spikes during the wind-driven downwelling and following relaxation period distinctly differ from the mean vertical thermohaline structure observed in the surrounding waters."

Line 213: "Can" was changed to "could".

Line 217: We changed "Also" to "But".

Line 221-223: We added one new sentence at the end of paragraph:

"All together, these inconsistencies inhibit linking the observed intrusions with interaction between the on-shelf warm halocline waters and the off-shelf cold Polar Water."

Line 224-226: We changed the first sentence in paragraph as follows:

"The alternative source of the observed thermohaline intrusions in front of glacier terminus was discussed in Dmitrenko et al. (2017). The authors suggested that relatively small cold intrusions of turbid water in the vicinity of the FIIC glacier observed at ~87 m could be attributed to subglacial waters enriched with suspended matter."

Line 231-234: We modified the beginning of paragraph as:

"In order to examine if the thermohaline properties within observed intrusions match those of subglacial water, we use the CTD data obtained over the ice-free area west of the FIIC glacier tongue in August 2015. These data showed the absence of warm halocline water in the vicinity of the glacier, and Bendtsen et al. (2017) argued that bottom water from the shelf can intrude below the tidewater glacier and cool to the freezing temperature through heat loss to the glacial ice."

Line 242-243: We changed the sentence as follows:

"Moreover, several CTD stations carried out within the debris of the glacier terminus in April 2016 (stations #26-28, Fig. 1b) show water temperatures between -1.67 and -1.74°C at the 33.2 isohaline **which also sustains the sub-glacial origin of observed cold intrusions in ITP records.**"

**Line 260-263: We changed first sentence to:**

"In previous section we suggested that cold sub-glacial water can be intruded into surrounding shelf waters as a result of frontal instability associated with the intensification of along-glacier currents during storms. In the following, we discuss another potential source of local instability related to the continual generation of internal tidal waves in front of the glacier. As already mentioned, tidal currents considerably contribute to the flow variability at the ITP site."

Line 328: The date was corrected to "August 2015"

**Other comments:**

- line 129 (p5): halostad? This is the first time I see this term.

The term halostad has first been used by Shimada et al. (2005) in his "Halocline structure in the Canada Basin of the Arctic Ocean" paper. He identified "halostad" as a layer with weak salinity stratification, relative to the adjoining high-gradient halocline layers, and temperature relatively close to the temperature of freezing. For the Wandel Sea, this term is used in paper by Dmitrenko et al. (2017) that is submitted to Ocean Science.

We slightly modified the text to clarify the meaning of halostad. The sentence in line 129 has been changed to: "Within the cold halostad (the layer of weakly stratified salinity below the surface mixed layer (Shimada et al., 2005))), represented by PcW, salinity increased from about 30 to 31.5 and temperature decreased from -1.6 °C to -1.7 °C (Fig. 2a,b)."

- line 238 (p7): why is there no basal melting? The intrusions sound less problematic then.

We agree with the reviewer that cooling of intruding water under the tidewater glacier is suggested to cause the basal melting and freshening of warm halocline. However, the basal melting would not occur, if the ice-ocean heat flux was relatively small and not exceeding the conductive heat flux to the glacier interior. Unfortunately, the lack of observational data beneath the glacier does not allow to confirm or contest the suggestion made in Bendtsen et al. (2017) paper.

- Figure 4, lower panels: have vertical lines over it to indicate the days, or at least the red and blue boxes of episodes A and B.

We have duplicated the red and blue boxes of episodes A and B in lower panel.

- Figure 6: in the caption, the distinction between what is measured from PS and from ITPs is unclear - everything looks like grey curves to me. Choose different symbols or a different wording.

We have decided to change the color of temperature, salinity, and TS profiles measured from Polarstern. In a new version of the figure, the magenta color is used to indicate the Polarstern data to avoid confusion with ITP scatters.

- Figure 7: ylabel of the insets is missing

Done as requested.

Regards,

On behalf of all authors

Sergei Kirillov

**Interactive comment by anonymous referee #2**

"I find the manuscript mostly clear and with interesting results. I have only one request; the color coding of DOY (Day Of Year) in Figure 6c is not visualizing the spikes in temperature very well. It would be nice if the authors could change that, perhaps emphasize the A and B storm events. This goes for the inserts in Figure 7 as well."

**Response to interactive comment of anonymous referee #2**

We highly appreciate the reviewer's comments and have tried to do the best to satisfy his/her request about the color scale in Fig. 6c and Fig. 7. However, the initially chosen pattern seems to represent one of the best visualization of data we could find. Unfortunately, using more saturated patterns (i.e. having wider color spectrum) sometimes results in very odd visual effects. The example of such color scale can be seen in figure 1.

The reason why it is difficult to find the optimal color scale is a large amount of vertical profiles presented in Fig. 6 and Fig. 7. It is nearly impossible to find the color scale that would be able to emphasize the different periods of the storm event. In fact, these figures aim to demonstrate the whole scope of TS or vertical profiles rather than the individual intrusions. The latter are shown separately in Fig. 8 where individual spikes in temperature are traced for different periods of storm.

Although we suggest that the initial color scale is more or less suitable, we have tried to find an alternative variant that would demonstrate the individual spikes more clearly. The figures with a new color scale are used in the new version of manuscript.

Regards,

On behalf of all authors

Sergei Kirillov